# Interplay between SERCA, 4E-BP, and eIF4E in the *Drosophila* heart

**Manuela Santalla**[1,2], **Alejandra García**[3], **Alicia Mattiazzi**[2], **Carlos A. Valverde**[2],
**Ronja Schiemann**[4], **Achim Paululat**[4], **Greco Hernández**[3], **Heiko Meyer**[4]*,
**Paola Ferrero**[1,2]*

**1** Departamento de Ciencias Básicas y Experimentales, UNNOBA, Pergamino, Buenos Aires, Argentina,
**2** Centro de Investigaciones Cardiovasculares 'Dr. Horacio E. Cingolani', CONICET-UNLP, La Plata, Buenos
Aires, Argentina, **3** Translation and Cancer Laboratory, Unit of Biomedical Research on Cancer, National
Institute of Cancer (Instituto Nacional de Cancerología, INCan), Mexico City, Mexico, **4** Department of
Zoology & Developmental Biology, Osnabrück University, Osnabrück, Germany

* pvferrero@comunidad.unnoba.edu.ar (PF); Meyer@biologie.uni-osnabrueck.de (HM)

Medical Research, AUSTRALIA

**Data Availability Statement:** All relevant data are
within the manuscript and its Supporting
Information files.

**Funding:** This work was supported by PICT 2014-
2549 ANPCyT and SIB 2015 to P.F, by PICT 2014-

## Abstract

Appropriate cardiac performance depends on a tightly controlled handling of $Ca^{2+}$ in a broad
range of species, from invertebrates to mammals. The role of the $Ca^{2+}$ ATPase, SERCA, in
$Ca^{2+}$ handling is pivotal, and its activity is regulated, inter alia, by interacting with distinct pro-
teins. Herein, we give evidence that 4E binding protein (4E-BP) is a novel regulator of
SERCA activity in *Drosophila melanogaster* during cardiac function. Flies over-expressing
4E-BP showed improved cardiac performance in young individuals associated with incre-
mented SERCA activity. Moreover, we demonstrate that SERCA interacts with translation
initiation factors eIF4E-1, eIF4E-2 and eIF4E-4 in a yeast two-hybrid assay. The specific
identification of eIF4E-4 in cardiac tissue leads us to propose that the interaction of eIF4E-4
with SERCA may be the basis of the cardiac effects observed in 4E-BP over-expressing
flies associated with incremented SERCA activity.

## Introduction

Eukaryotic translation initiation factor 4E (eIF4E) and eIF4E-binding proteins (4E-BPs) play
crucial roles in mRNA cap-dependent translation. This mechanism begins with the recogni-
tion of the cap structure ($m^7$GpppN, where N is any nucleotide) located at the 5' end of the
mRNA by eIF4E, which forms a complex with eIF4G. Formation of the eIF4E–eIF4G complex
is one of the most critical events regulating the global process of cap-dependent translation.
Therefore, eIF4E represents a major target for gene expression regulation [1]. eIF4E is regu-
lated in mammals by three 4E-BP paralogs (1, 2 and 3) that share the eIF4E-binding motif
YXXXXLφ with eIF4G (where X is any amino acid, and φ is a hydrophobic residue); the latter
interacts with the dorsal surface of eIF4E [2]. Thus, binding of 4E-BPs to eIF4E blocks its asso-
ciation with eIF4G, thereby repressing cap-dependent translation. 4E-BPs are phosphorylated
by the TOR kinase downstream of the PI3K signalling pathway [3]. Activity of 4E-BPs is mod-
ulated by the phosphorylation status; whereas hypo-phosphorylated 4E-BPs show high affinity
for eIF4E, inhibiting translation, hyper-phosphorylated species dissociate from eIF4E, thus

2524 and PIP 0350 to AM and by grants from the Deutsche Forschungsgemeinschaft to A.P. and H. M (SFB 944, Physiology and dynamics of cellular microcompartments). G.H. was supported by internal funding of the National Institute of Cancer (Instituto Nacional de Cancerología, INCan), Mexico. A National Council of Science and Technology (CONACyT) PhD fellowship was awarded to A.G. (Nr. 436200). A.G. is part of the Program on Biological Sciences, UNAM (Doctorado en Ciencias Biológicas de la Universidad Nacional Autónoma de México).The funders had no role in study design, data collection and analysis, decision to publish, or preparation of the manuscript.

**Competing interests:** The authors have declared that no competing interests exist.

promoting protein synthesis [4]. Hence, several phenomena depending on translation are affected by 4E-BP activity and its interactors. However, other actions of 4E-BP—beyond protein synthesis—are not completely understood.

*Drosophila melanogaster* harbours one single gene of 4E-BP, termed *d4E-BP* or *Thor*, one gene of TOR kinase, termed *dTOR*, and seven genes encoding eIF4E proteins [5–8]. eIF4E 1–2 isoforms are result of alternative splicing from the same gene [5]. As mentioned previously, 4E-BP can bind to the main eIF4E protein participating in the translation process—eIF4E-1— but it also binds to other eIF4E proteins present in *Drosophila* [5]. In a physiological context and focusing on cardiac function, it has been observed that *d4E-BP* protects against the deleterious effects of aging, whereas up-regulation of eIF4E-1 impairs cardiac function even in young adults [9]. In this seminal work, the impairment of cardiac function was assessed by the susceptibility of flies to either fibrillation or cardiac arrest when challenged with a specific stress (i.e., an increase in stimulation frequency). This abnormal response to stress was termed *heart failure* [9].

In *Drosophila* and mammals, the cycle of each heartbeat is the result of the cardiomyocyte response to changes in membrane potential and intracellular $Ca^{2+}$ concentrations $[Ca^{2+}]$. The mechanism by which the electrical impulse (action potential) leads to muscle contraction is termed excitation–contraction coupling (ECC). After membrane depolarisation, $Ca^{2+}$ enters the cytosol through L-type $Ca^{+2}$ channels. $Ca^{2+}$ stimulates the ryanodine receptors (RyR) in the sarcoplasmic reticulum (SR) membrane, promoting the release of $Ca^{2+}$ from this reservoir [10]. The overall increase in cytosolic $[Ca^{2+}]$ induces contraction, which is carried out by $Ca^{2+}$-dependent steric modulation of myosin cross-bridge cycling on actin by the thin filament troponin-tropomyosin complex. $Ca^{2+}$ is then transported back into the SR via the $Ca^{2+}$-ATPase SERCA and extruded from the cell through the $Na^+/Ca^{2+}$ exchanger (NCX) in the plasma membrane to initiate relaxation. The increase and subsequent decrease in intracellular $[Ca^{2+}]$ is referred to as *$Ca^{2+}$ transient* [11]. *Drosophila* heart modulation involves several cardiac proteins and regulators like neuropeptides and peptide hormones that significantly affect heart parameters [12].

In flies, aging results in a reduction in spontaneous cardiac frequency and modifies intracellular $Ca^{2+}$ dynamics by slowing cardiac relaxation. Moreover, frequency variability increases in older flies [13, 14]. As previously mentioned, eIF4E and 4E-BP participate in cardiac aging of *Drosophila melanogaster* [8]. However, the impact of these proteins on mechanisms involved in cardiac function at the molecular level (e.g., the $Ca^{2+}$ handling, which constitutes the basic mechanism responsible for cardiac contractility and function) has not been explored.

The aim of the present work was to study the role of *d4E-BP* on cardiac $Ca^{2+}$ handling in non-stressed hearts. Moreover, our research provides insights into the function of eIF4E-4 protein in cardiac tissue and its molecular association with SERCA using *Drosophila melanogaster* as a model.

## Materials and methods

### *Drosophila* strains and fly work

The transgenic flies over-expressing *d4E-BP* (*w\*;P[UAS-Thor.wt]2*) [7], *dTOR* (*y[1]w[\*]Pry[+t7.2] = hsFLP-12; P-w[+mC] = UAS-Tor.WT-III*) [15] and the wildtype *Canton S* were obtained from Bloomington *Drosophila* Stock Center, Indiana University, USA (BDSC_9147, BDSC_7012 and BDSC_64349 respectively). The driver fly strain *TinC-Gal4-UAS-GCaMP3* was a kind gift from Matthew Wolf (University of Virginia, USA) [16].

All stocks were maintained and amplified at 25˚C on standard cornmeal-yeast medium. Transgenic flies that over-expressed 4E-BP and TOR under UAS were crossed with flies

harbouring the heart-specific reporter system GCaMP3 and the Gal4 activator under the control of *tinC* promoter. This promoter belongs to a transcription factor present in cardiac cells only [17]. Heterozygous individuals of F1 progeny were grown at 28˚C until 7 and 40 days of adult life. Progeny resulting from Canton-S crossed with reporter construct was used as control.

The life span of fruit flies is approximately 50-days in the lab [18]. Forty-day-old flies were chosen because such individuals are mature adults, and manipulation of them involves fewer difficulties than 60-day-old flies. Comparing our own experiments, 40 versus 60-day-old flies exhibit a similar deterioration in cardiac performance. Moreover, according to previous reports from other groups, 5-week-old flies manifest symptoms of aging [9]. During aging, F1 individuals were raised in vials with meal and changed to new vials every 5 days to avoid contamination with following offspring.

## Heart dissection

Flies collected 7 and 40 days after hatching were used for cardiac $Ca^{2+}$ fluorescence measurements. Hearts were dissected and prepared as described [14]. Adult flies were anaesthetised with $CO_2$, and thorax and legs were removed from the fly. Flies were bathed in oxygenated haemolymph buffer (108 mM NaCl, 5 mM KCl, 8 mM $MgCl_2$, 1 mM $NaH_2PO_4$, 4 mM $NaHCO_3$, 5 mM HEPES pH 7.1, 10 mM sucrose, 5 mM trehalose and 2 mM $CaCl_2$) [19] at room temperature. The semi-intact preparation was mounted in a confocal microscope (Carl Zeiss 410).

## Functional analysis

Beating hearts from semi-intact preparations were observed, and changes in GCaMP3 fluorescence in the first chamber were recorded (conical chamber). Transient elevation of cytosolic $Ca^{2+}$ concentration that precedes cardiac contraction was detected as a fluorescent signal. Recording and parameter measurements were carried out as described in Santalla et al. 2014 [14]. The $Ca^{2+}$ transients were recorded during 30 seconds resulting in a 1024-pixel image. The images obtained were interpreted to graph sequentially in time the intensity of fluorescence (S1 Fig), mediating a customized algorithm using the Jupiter notebook program, based on Python programming language, with Matplotlib and NumPy libraries from Anaconda [20]. Values expressed as relative change of fluorescence along time, calculated according to the formula: Fmax-F0/F0 and expressed in arbitrary units of fluorescence (AU), were analyzed with LabChart software (AD Instruments, CO, USA). Measurements included: peak $Ca^{2+}$ transient amplitude (Fmax-F0/F0) (AU), maximal rates of fluorescence increase and decrease (+ΔF/dt) (AU/sec), (-ΔF/dt) (AU/sec). Relaxation was measured by calculating the Tau constant (in seconds) of the transient exponential decay of $Ca^{2+}$ (S1 Fig). For estimating SERCA activity, a caffeine pulse (10 mM) was applied in the perfusion media after 25 seconds of fluorescence recording. This compound releases calcium content through RyR [21]. SERCA activity was estimated by subtracting the inverse of the rate constant of decay of the caffeine-evoked transient (Taucaff) from that of the systolic $Ca^{2+}$ transient (1/TauCa) (1/Taucaff—1/TauCa). This calculation assumes that the decay of the systolic $Ca^{2+}$ transient is produced by SERCA and NCX activity, whereas caffeine-induced $Ca^{2+}$ transient decay is produced by NCX activity only [20]. Analyzed data sets are presented in S2 Table.

## Construction of plasmids and two-hybrid assay

**Plasmids.** *Drosophila* 4E-BP (*Thor*) cDNA [6] and SERCA (CG3725) amino acids 1–250, 51–250 and 200–250 were cloned as Ncol-Pstl fragments onto the vector pOAD ('prey'; [22])

in-frame with the activator domain (AD) sequence of GAL4 to generate the constructs p4E-BP-AD and pSERCA(1–250)-AD, pSERCA(51–250)-AD and pSERCA(200–250)-AD respectively. *Drosophila* eIF4E cognate cDNAs [5] were cloned into the pOBD2 vector ('bait'; [22]) in frame with the DNA-binding domain (BD) sequence of GAL4 to create the respective plasmids peIF4Es-BD.

**Yeast two-hybrid system.**   Interactions between 'bait' and 'prey' proteins were detected following a yeast interaction-mating method using the strains PJ69-4a and PJ69-4alpha [22]. Diploid cells containing both bait and prey plasmids were grown in selective media -(Trp, Leu) as growth control. Protein interactions were detected by replica-plating diploid cells onto selective media -(Trp, Leu, His) + 3 mM, 5 mM, 10 mM, 20 mM or 30 mM 3-amino-1,2,4-tria-zole (3AT). Growth was scored after 4 days of growth at 30˚C.

## Mass spectrometry analysis

Three-hundred hearts were isolated and processed in RIPA buffer, sonicating them for 30 sec twice at 65 W/L. Fifteen μg of proteins were subjected to SDS-PAGE and further mass spectrometry analyses. Protein digestion and mass spectrometry analysis were performed at the Proteomics Core Facility CEQUIBIEM, at the University of Buenos Aires/CONICET (National Research Council) as follows: Coomassie-stained SDS-PAGE gel excised protein bands were sequentially washed and destained with 50 mM ammonium bicarbonate, 25 mM ammonium bicarbonate 50% acetonitrile and 100% acetonitrile; reduced and alkylated with 10 mM DTT and 20 mM iodoacetamide and in-gel digested with 100 ng Trypsin (Promega V5111) in 25 mM ammonium bicarbonate overnight at 37˚C [23].

Peptides were recovered by elution with 50% acetonitrile-0.5% trifluoroacetic acid, including brief sonication, and further concentrated by speed-vacuum drying.

Samples were resuspended in 15 μl of water containing 0.1% formic acid, desalted using C18 zip tips (Merck Millipore) and eluted in 10 μl of $H_2O$:ACN:FA 40:60:0.1%.

Then, the peptides were purified and desalted with ZipTip C18 columns (Millipore). The digests were analyzed by nanoLC-MS/MS in a Thermo Scientific QExactive Mass Spectrometer coupled to a nanoHPLC EASY-nLC 1000 (Thermo Scientific). For the LC-MS/MS analysis, approximately 1 μg of peptides was loaded onto the column and eluted for 120 minutes using a reverse phase column (C18, 2 μm, 100A, 50 μm x 150 mm) Easy-Spray Column PepMap RSLC (P/N ES801) suitable for separating protein complexes with a high degree of resolution. The flow rate used for the nano column was 300 nL min-1 with a solvent range from 7% B (5 min) to 35% (120 min). Solvent A was 0.1% formic acid in water, whereas B was 0.1% formic acid in acetonitrile. The injection volume was 2 μL. The MS equipment has a high collision dissociation cell (HCD) for fragmentation and an Orbitrap analyser (Thermo Scientific, Q-Exactive). A voltage of 3.5 kV was used for electrospray ionisation (Thermo Scientific, EASY-SPRAY) [23].

XCalibur 3.0.63 (Thermo Scientific) software was used for data acquisition and equipment configuration that allows peptide identification at the same time of their chromatographic separation.

Data-dependent acquisition was performed with the following method: Full-scan mass spectra were acquired in the Orbitrap analyser. The scanned mass range was 400–1800 m/z, at a resolution of 70000 at 400 m/z, and the 12 most intense ions in each cycle were sequentially isolated, fragmented by HCD and measured in the Orbitrap analyser. Peptides with a charge of +1 or with unassigned charge state were excluded from fragmentation for MS2 [23].

A targeted acquisition method was also performed. A list with the m/z values resulting from the theoretical digestion of eIF4E protein and of previously detected eIF4E m/z values

(www.peptideatlas.org) was added to the XCalibur method, so as to ensure that the m/z values present in the list were selected for fragmentation. This method favours the detection of the peptides of the protein of interest, regardless of their relative abundance in the sample.

## Analysis of MS data

We obtained raw data, which was processed using Proteome Discoverer software (version 2.1.1.21 Thermo Scientific). Then, we carried out a search against the *Drosophila melanogaster* sequences database. Protein hits were filtered for high confidence peptide matches with a maximum protein and peptide false discovery rate of 1% calculated by employing a reverse database strategy.

## RT-PCR of *Drosophila* heart

To evaluate the expression levels of eIF4E-4, 70 hearts were isolated, and RNA was extracted by the TRIzol method (Thermo Fisher # 15596026) according to the manufacturer's instructions. Then, PCR amplification was performed with the OneStep RT-PCR kit from Qiagen. This kit has both a retrotranscriptase and DNA polymerase, allowing the retrotranscription of the RNA and subsequent amplification of DNA in a single step.

The primers used were:

eIF4E-4 Fwd: 5 'CCGTTATCAACTTGCGCGG 3'
eIF4E-4 Rev: 5 'CCCTGCTTGCACATAGTGTC 3'
Tubulin Fwd: 5 'ATCAACTACCAGCCTCCCAC 3'
Tubulin Rev: 5 'TCCTCCATCCCCTCCCCAAC 3'

## Statistical analysis

All results from functional analyses of calcium handling were expressed as mean ± SEM. Comparisons were made using one or two-way ANOVA with a Tukey test when comparing the effect of lines or age and lines, respectively. P-values $< 0.05$ were considered to be statistically significant.

# Results

## 4E-BP is involved in cardiac $Ca^{2+}$ handling

First, we analyzed the effect of the 4E-BP cardiac-specific overexpression [6] on the background of *tinC-Gal4*, *UAS-GCaMP3* [16] on heart rate and intracellular $Ca^{2+}$ dynamics in 7- and 40-day-old flies. Previous experiments indicated that 60-day-old flies display a decrease in heart rate, associated with a slowing of relaxation [14]. We detected similar effects of aging on heart rate (Fig 1A) and Tau relaxation constant (Fig 1E) in 40-day-old flies. Seven-day-old flies over-expressing 4E-BP exhibited an increase in the amplitude of the intracellular $Ca^{2+}$ transient (Fig 1B). This effect was accompanied by an increase in the rates of contraction (positive derivative) and relaxation (negative derivative) (Fig 1C and 1D), indicating an influence of 4E-BP overexpression on cardiac contractility. 4E-BP overexpression also modified all parameters in a similar manner in old flies, although the effects were less pronounced than in young animals and did not reach statistical significance. A possible explanation for these results may be that the beneficial effect of 4E-BP overexpression is not sustained and might be countered in older flies by the concurrent deleterious influence of aging.

To assess the underlying mechanism associated with $Ca^{2+}$ handling, we analyzed SERCA activity. SERCA protein is a $Ca^{2+}$ ATPase in the membrane of the SR within cardiac cells, responsible for $Ca^{2+}$ uptake into the SR, thus contributing to muscle relaxation [19, 23]. In

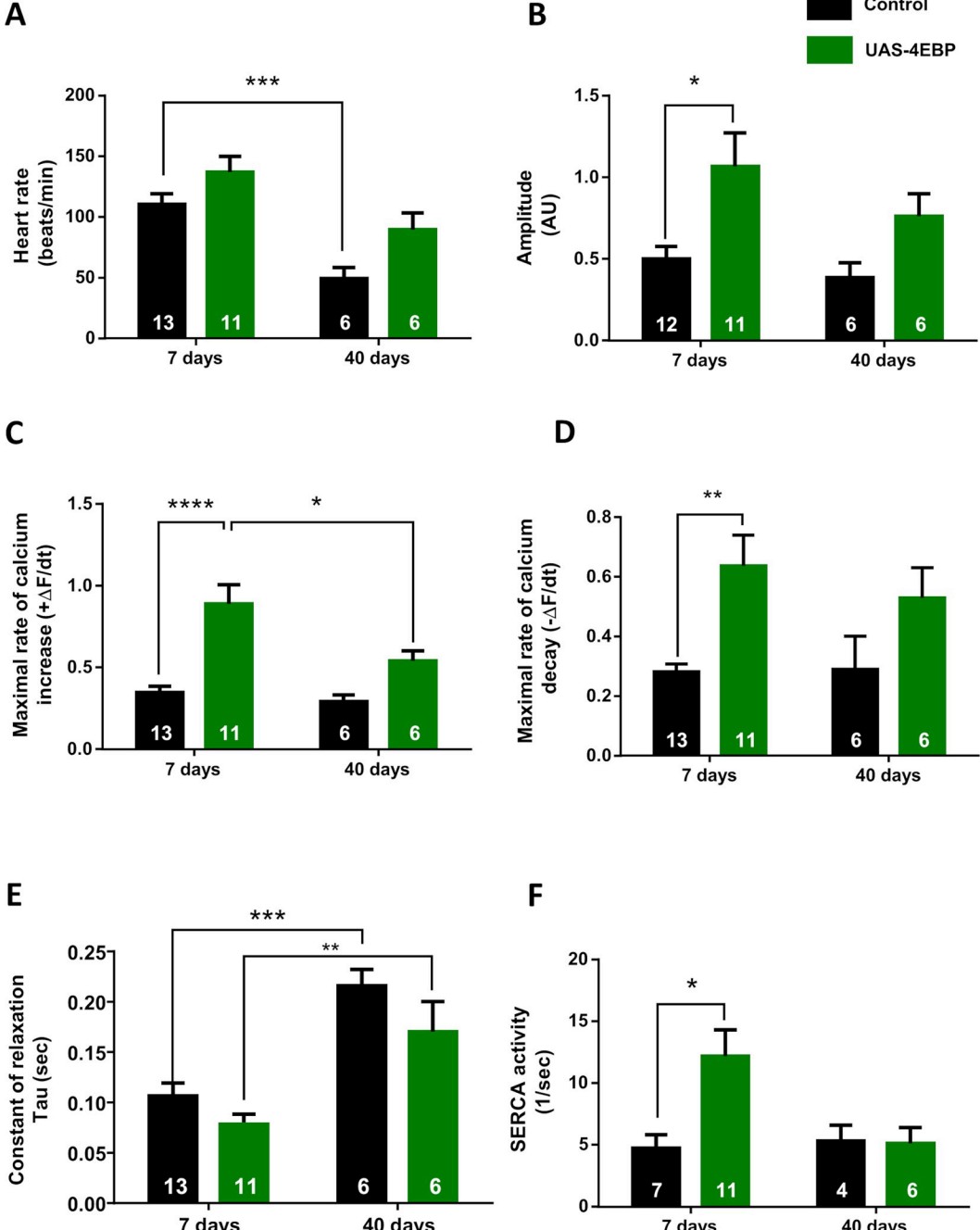

**Fig 1. Overexpression of 4E-BP improves cardiac performance.** Average values of intracellular $Ca^{2+}$ cycling parameters. *Black bars*, control flies (*tinC*-Gal4, UAS-GCaMP3/+). *Green bars*, flies overexpressing 4E-BP in cardiac tissue (*tinC*-Gal4, UAS-GCaMP3 / UAS-4E-BP). Overexpression of 4E-BP did not produce significant changes in heart rate (A). Instead, significantly increased $Ca^{2+}$ transient amplitude (B), maximum rate of $Ca^{2+}$ transient increase ($\Delta F/dt_{max}$) and decay ($\Delta F/dt_{min}$) (C–D) were observed in 7-day-old flies. Time constant of $Ca^{2+}$ transient decay—Tau (E)—was not significantly changed, whereas the estimated SERCA activity (F) was incremented. Overexpression of 4E-BP in 40-day-old flies did not increase these parameters significantly, compared to control flies. All results are expressed as mean ± SEM. * $p < 0.05$, ** $p < 0.01$ and **** $p < 0.001$.

mammals, SERCA2A is the cardiac variant, whereas *Drosophila* has only one gene encoding the SERCA protein [20, 24]. SERCA activity was estimated by the difference between the decay of the $Ca^{2+}$ transient, compared to the decay of the caffeine-induced $Ca^{2+}$ release with caffeine being applied onto semi-intact heart preparations. Caffeine, opens the ryanodine receptors and releases most of the $Ca^{2+}$ stored inside the SR which is extruded from the cell throughout the NCX. It has been shown that the decay of the systolic $Ca^{2+}$ transient reflects the combined activity of SERCA and NCX, whereas caffeine-induced $Ca^{2+}$ transient decay depends exclusively on the NCX activity [21] (See Material and Methods for details). In flies over-expressing 4E-BP, we found an increase in SERCA activity at 7 days (Fig 1F), concurrent with the tendency of a decrease in Tau relaxation constant and a significant increase in the maximal velocity of relaxation observed in these young flies (Fig 1E). To verify a direct impact of SERCA activity on calcium transient decay, we analyzed the constant of relaxation Tau for a subset of paired data in which SERCA activities was measured. An increase in SERCA, were correlated with an acceleration of relaxation (S1 Table).

In contrast, overexpression of 4E-BP did not significantly increase either SERCA activity or the relaxation velocity in 40-day-old flies, compared to control animals (Fig 1F). Together, our results support a role of 4E-BP in regulating cardiac performance and SERCA activity predominantly in young age.

## Intracellular $Ca^{2+}$ transient is not affected by augmented TOR expression, but it is altered during starvation

As discussed previously, 4E-BP activity is associated with dTOR signalling. A key function of 4E-BP is eIF4E binding and inhibition. This process is regulated by dTOR-dependent phosphorylation of 4E-BP, which leads to 4E-BP inactivation [25]. Therefore, we analyzed cardiac performance in animals, in which dTOR was over-expressed. In addition, we implemented starvation conditions into our cardiac performance analysis because it has been shown that starvation results in reduced dTOR signalling [26]. For starvation, five-day-old animals were kept for additional 48 hours on 4% agar (in water) without food. Results were compared to those obtained from animals not exposed to starvation.

Fig 2 shows that 7-day-old flies overexpressing dTOR exhibited similar values of heart rate (A), amplitude (B), rates of $Ca^{2+}$ transient increase and decay (C, D), time constant of relaxation (E) and SERCA activity (F), compared to control flies. However, the starvation induced an increase in heart rate, accompanied by reduced amplitude, most likely due to augmented velocities of $Ca^{2+}$ transient development (Fig 2A–2D). Although the time constant of relaxation was not significantly altered (Fig 2E), strong SERCA activation was detected (Fig 2F). Thus, although dTOR overexpression seems not to contribute significantly to cardiac $Ca^{2+}$ handling, its presumable inhibition partially modifies cardiac performance. This result however has to be considered with caution because starvation certainly has an impact on dTOR in numerous tissues in addition to the heart (e.g., the fat body of *Drosophila melanogaster*). Moreover, 4E-BP levels might be increased through FOXO regulation via the Akt pathway [27].

Next, we analyzed whether starvation augmented the cardiac phenotype induced by 4E-BP overexpression in the corresponding flies. An increment in heart rate was observed in starved 4E-BP flies (Fig 3A). However, this challenge did not exacerbate the effects of 4E-BP overexpression on $Ca^{+2}$ transient characteristics (Fig 3B–3F).

## SERCA interacts with eIF4E-4

Increased levels of 4E-BP improve cardiac function; in particular, the activity of SERCA is significantly increased. Based on this result, we tested the hypothesis that the increase in activity

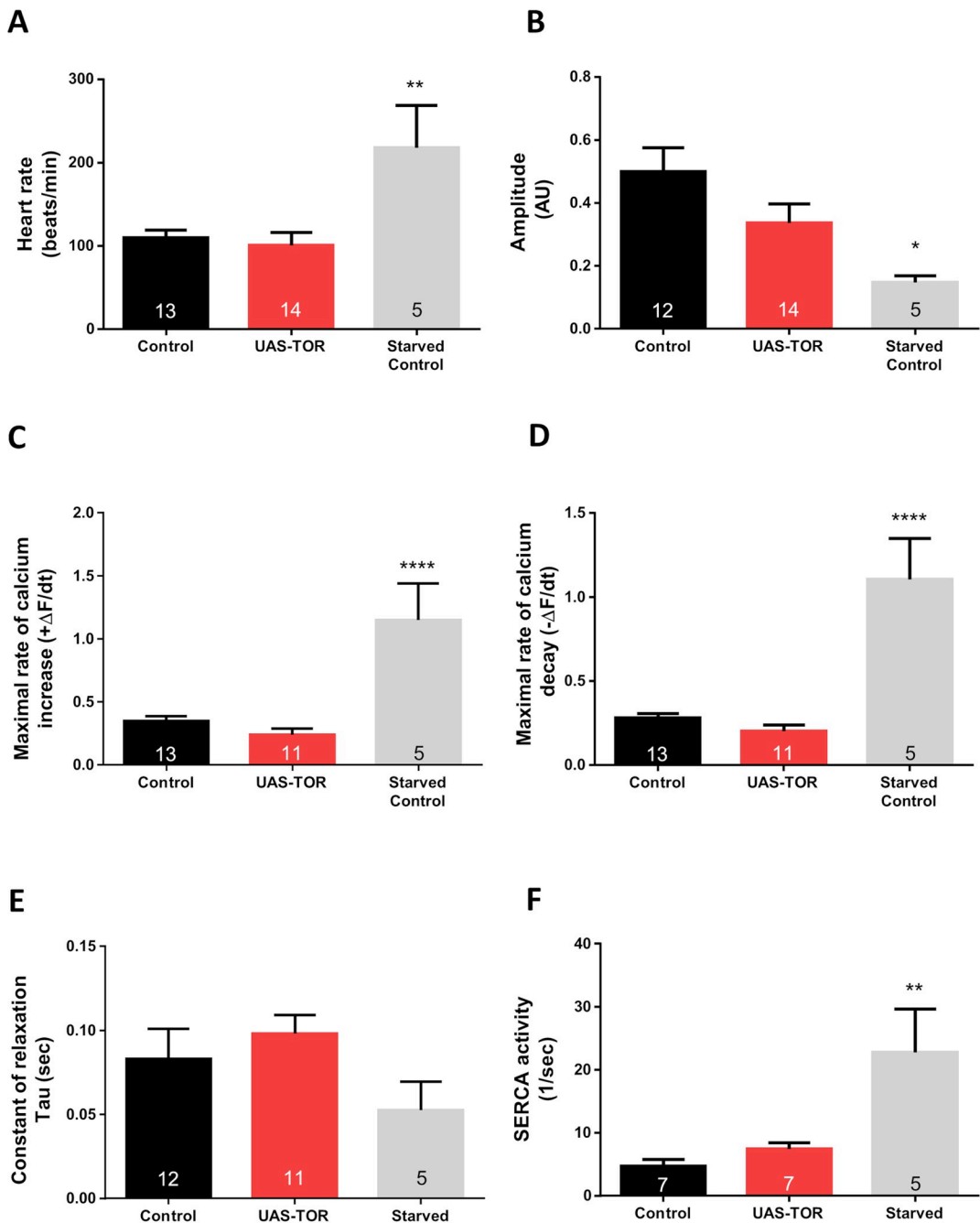

**Fig 2. Intracellular Ca²⁺ transient is not affected by augmented dTOR expression, but it is altered during starvation in young wild type flies.** Tor overexpression did not change heart rate (A), Ca²⁺ transient amplitude (B), maximum rate of Ca²⁺ transient increase ($\Delta F/dt_{max}$) and decay ($\Delta F/dt_{min}$) (C–D), time constant of Ca²⁺ transient decay, Tau (E) or SERCA activity (F) in 7-day-old flies. Instead, 48 hr of starvation drastically accelerated maximum rate of Ca²⁺ transient increase ($\Delta F/dt_{max}$) and decay ($\Delta F/dt_{min}$) (C–D) that provoked reduction in the Ca²⁺ transient amplitude (B) probably due to incremented heart rate (A) in control flies (*tinC*-Gal4, UAS-GCaMP3 / +). These changes were accompanied by augmented SERCA activity. All results are expressed as mean ± SEM. * $p<0.05$, ** $p<0.01$ and **** $p< 0.001$.

is due to an interaction between SERCA and eIF4E, which could be modulated by 4E-BP. To determine whether SERCA binds to any of the eIF4E cognates [5], yeast two-hybrid assays were performed using SERCA as 'prey' and the different eIF4E proteins as"bait"' (Fig 4A). A

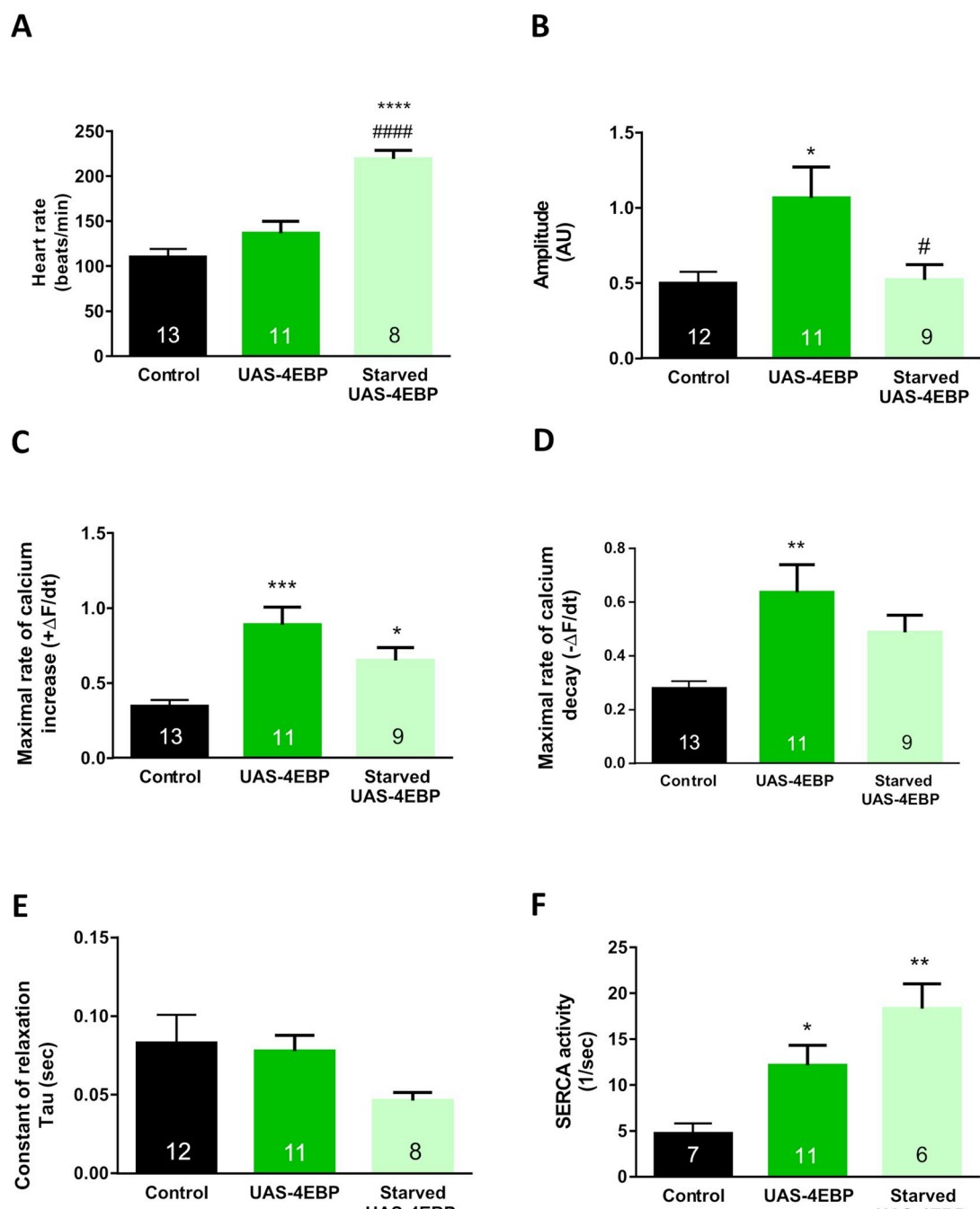

**Fig 3. Starvation alters intracellular Ca$^{2+}$ transient in 4E-BP OE flies.** Forty-eight hours of starvation increased heart rate (A) in 7-day-old 4E-BP overexpression flies (A) and decreased the amplitude of the Ca$^{2+}$ transient (B). Maximal rates of Ca$^{2+}$ transient increase ($\Delta F/dt_{max}$) and decay ($\Delta fFdt_{min}$) (C–D) were reduced. The time constant of Ca$^{2+}$ transient decay—Tau (E)— remained without changes (E). SERCA activity was augmented (F) compared to control non-starved flies. All results are expressed as mean ± SEM. * $p<0.05$ with respect to control group; *** $p<0.001$ **** $p<0.0001$; # $p<0.05$ with respect to UAS-4E-BP flies, #### $p<0.0001$.

strong interaction was detected between SERCA and eIF4E-4, as well as weaker interactions between SERCA and eIF4E-1 and eIF4E-2. eIF4E-3 was not tested as this variant is only expressed in testes [28]. eIF4E-5 was not tested either because it gives a high growth

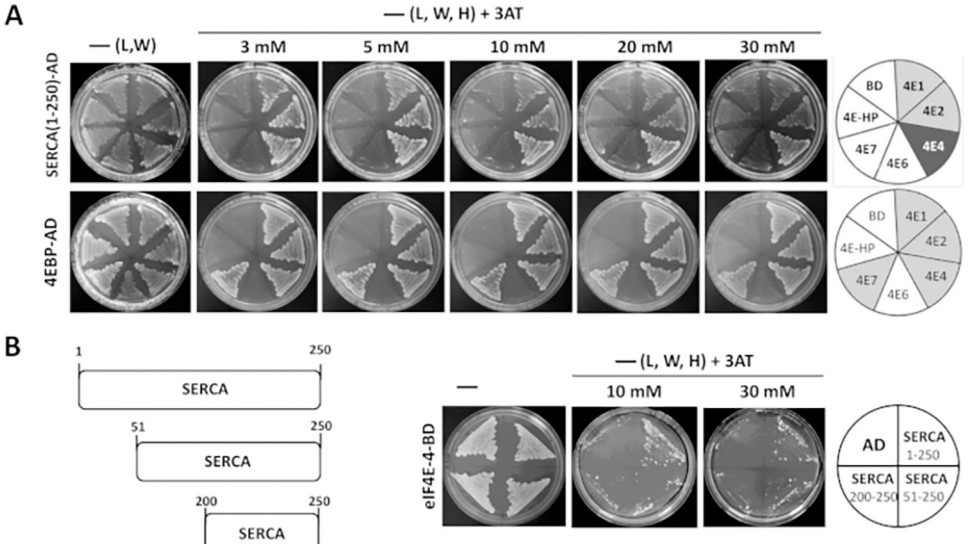

**Fig 4. SERCA interacts with eIF4E-4, eIF4E-1 and eIF4E-2 in the yeast two-hybrid system.** A) "Prey" SERCA (amino acids 1–250)-activator domain (AD; upper panel) interacts with "baits" eIF4E-4, eIF4E-1 and eIF4E-2 in a yeast two-hybrid assay. eIF4E-4 exhibits the strongest interaction. "Prey" 4E-BP-AD (lower panel) was used as a positive control. Empty vector (BD) was used as a negative control. B) pSERCA(51–250)-AD, but not pSERCA(200–250)-AD, interacts with eIF4E-4-BD. L, leucine; W, tryptophan; H, histidine; 3AT, 3-amino-1,2,4-triazole.–(L,W), growth control. Interactions were tested under increasing astringency conditions (3 to 30 mM 3-amino-1,2,4-triazole, 3AT).

background by itself. 4E-BP was used as a positive control for interaction. To narrow down the SERCA domain interacting with eIF4E-4, which is the variant that showed the strongest interaction, we tested two shorter fragments of SERCA, namely amino acids 51–250 and 200–250. As shown in Fig 4B, the SERCA constructs 1–250 and 51–250, but not construct 200–250, interacted with eIF4E-4, suggesting that the SERCA eIF4E-binding motif is located within the region encompassing amino acids 1–200. A close inspection of the SERCA primary sequence did not identify a canonical eIF4E binding motif (YXXXXφ, where φ is any hydrophobic amino acid). However, as also described for other 4E interaction proteins (4E-IPs) [29], eIF4E binding may occur via non-canonical binding motifs, which might be present in the SERCA primary sequence.

## eIF4E-4 is expressed in cardiac tissue of *Drosophila* melanogaster

eIF4E-1 is referred to as the ubiquitous eIF4E variant. eIF4E-1 is expressed in all tissues of *Drosophila melanogaster* throughout development [5, 26]. In contrast, expression of the other eIF4E proteins has been poorly characterised, and the respective physiological roles are largely unclear [5]. Given that our results indicate an interaction between eIF4E-4 and SERCA, we wondered whether this variant is expressed in cardiac tissue of adult flies, which would support a possible function in heart physiology. Therefore, we evaluated and confirmed mRNA expression of *eIF4E-4* by RT-PCR analysis of isolated cardiac tissue (Fig 5 and S1 Raw image).

In addition to the mRNA detection, we also aimed to confirm presence of the eIF4E-4 protein in heart cells. Due to a lack of antibodies that specifically detect eIF4E-4, we assessed protein presence via a mass spectrometry analysis of isolated *Drosophila melanogaster* hearts. Comparison of the peptide profiles obtained by targeted mass spectrometry to the *Drosophila melanogaster* proteome database (Fruit fly- UP000000803) revealed that eIF4E-4 is present in cardiac tissue of *D. melanogaster* (Table 1). Two peptides (one with high confidence and one with medium confidence) indicated presence of the protein.

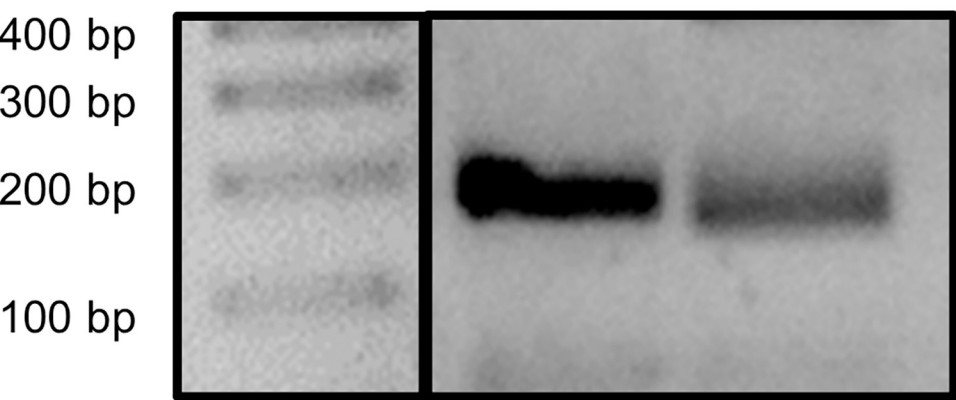

**Fig 5. mRNA of *eIF4E-4* is expressed in cardiac tissue of *Drosophila melanogaster*.** RT-PCR using specific primers to amplify a fragment of *tubulin* and of *eIF4E-4* was carried out using isolated cardiac tissue. The size of each respective amplicon corresponds to the expected size (*tubulin*: 182 bp; *eIF4E-4*: 166 bp).

## Discussion

4E-BP is one of the downstream effectors of PI3K/TOR signalling [30]. Its canonical role has been identified as a regulator of protein synthesis in eukaryotes because it interacts with eIF4E, a factor that promotes cap-dependent translation. 4E-BP sequesters eIF4E, precluding its function [31]. Pivotal findings of Wessells and co-workers [9] provided strong evidence regarding the essential role of both components of dTOR signalling in heart responses to stress during aging. Our findings support the relevance of 4E-BP observed by Wessells et al, in *d4eBP* null mutant flies, which exhibit a high failure rate at young age, whereas cardiac-specific overexpression of *d4eBP* was shown to be cardio protective against stress-induced failure [9]. In contrast to that work, herein, we analyzed the cardiac performance under basal conditions focusing on the underlying mechanisms of contraction, which involve $Ca^{2+}$ transient dynamics.

Our results demonstrate that overexpression of 4E-BP improves cardiac contractility through physiological mechanisms that regulate the cycling of $Ca^{2+}$. In young flies, an increase

**Table 1. Fragment of the results table generated by proteome discover with the list of identified proteins.** eIF4E-4 protein hit is shown with the correspondingly matching peptides indicated.

| Accession | Description | Coverage | # Peptides | # PSMs | # Unique Peptides | # AAs | MW [kDa] | calc. pI | Score Sequest HT |
|---|---|---|---|---|---|---|---|---|---|
| Q9VRY0 | GH23527p OS = *Drosophila melanogaster* GN = eIF4E-4 PE = 2 SV = 1 | 8,29694323 | 2 | 3 | 2 | 229 | 26,4 | 5,16 | 3,78 |
| | Confidence | Sequence | Modifications | # PSMs | # Missed Cleavages | Theo. MH+ [Da] | XCorr Sequest HT | Charge Sequest HT | m/z [Da] Sequest HT |
| | High | WVINMGRGSKAELDK | 1xOxidation [M5] | 2 | 2 | 1719,88467 | 2,09 | 2 | 860,44843 |
| | Medium | FGGRWVINMGR | | 1 | 1 | 1292,66807 | 1,28 | 2 | 646,83936 |

in the amplitude of the $Ca^{2+}$ transient was accompanied by increased rates of calcium increase and decay (Fig 1B–1D). The molecular phenomenon underlying the positive lusitropic effect of 4E-BP appears to be an increment of SERCA activity that can explain the enhancement of the $Ca^{2+}$ transient amplitude and the $Ca^{2+}$ transient decay rate. SERCA is responsible for the uptake of $Ca^{2+}$ into the SR and thereby essential for heart function. In mammals, a reduced SERCA2A activity, the cardiac specific variant of SERCA, causes heart failure [32–34]. In the *Drosophila* heart, SERCA activity is crucial for regulating the Bowditch effect, as shown by us in a previous work [20]. Interestingly, García-Casas et al. found that inhibition of SERCA extended the longevity of *C. elegans*, an animal that lacks a heart, indicating that $Ca^{2+}$ signalling plays a relevant role in the aging process. This observation highlights the relevance of SERCA to other processes related to $Ca^{2+}$ handling in different contexts and tissues.

Of note, although the augmented SERCA activity accelerated the calcium transient decay in young flies overexpressing 4E-PB (Fig 1D and 1F), the constant of relaxation Tau did not reach a significant reduction in corresponding individuals (Fig 1E). Therefore, an action of 4E-BP on other targets associated with intracellular calcium handling could not be ruled out. For example, it has been shown that 4E-BP modulates mitochondrial biogenesis and activity in cultured cells [35]. Mitochondria are a slow calcium removal system in vertebrates [36], but their contribution to intracellular calcium removal in the *Drosophila* heart is unknown.

Moreover, intracellular calcium handling is influenced by the sensitivity of contractile proteins to calcium. During the relaxation phase, calcium must be released from troponin C to promote dissociation of actin and myosin filaments. When the calcium sensitivity of contractile proteins is incremented, contractility also increases, but relaxation may be prolonged because calcium dissociates more slowly from troponin C [37]. Although we did not assess this aspect, changes in the sensitivity of contractile proteins to $Ca^{2+}$ could counteract the relaxant effect produced by increased SERCA activity in the analyzed individuals (Fig 1E).

In our analysis of the heart function, the beneficial effects of 4E-BP overexpression were less pronounced in old flies. Previous work from our laboratory indicated that aging reduces heart rate and prolongs relaxation time [14, 38]. Overexpression of 4E-BP in older flies induced a slight improvement of the parameters, compared to control flies of the same age. However, the effects were not sufficiently strong to compensate for the deleterious impact of aging. The estimated SERCA activity in old 4E-BP over-expressing flies was similar to the age-matched control group.

Another possible explanation for a less pronounced beneficial effect of 4E-BP overexpression in old flies is a declined activity of the Gal4 driver line. However, we see a constantly high signal of the CGaMP3 GFP reporter when driven with *tin*C-Gal4, rendering a weakened expression in old flies unlikely. During aging, the proteome stability changes differently in several organisms among vertebrates and invertebrates. For example, some studies reported a decrease in global proteome turnover in aged adult nematodes, particularly affecting proteins in microtubules, vitellogenins, translation (e.g., ribosomes) and mitochondria [38]. By contrast, the mouse proteome exhibits little to no overall global changes. For individual proteins, changes in their age-dependent turnover might vary, depending on the tissue, and levels of protein related to mitochondrial dysfunction, branched-chain amino acid metabolism, actin cytoskeleton or oxidative stress response were enriched in the heart [38]. Nevertheless, other proteins might not change. In rats, although the activity of SERCA2a decreased with age, no differences in relative protein amounts were observed [32]. Considering the difference in SERCA activity between young and old 4E-BP overexpressing flies but not between young and old control animals, further studies measuring the expression levels or posttranslational modifications of SERCA, with and without increased 4E-BP expression, might clarify this aspect.

Canonical 4E-BP function is regulated by the dAkt and dTOR pathways [27]. Normally, 4E-BP is directly inhibited by TOR kinase-mediated phosphorylation. Inhibition of TOR leads to an increase in the amount of hypo-phosphorylated 4E-BP in the cytosol [27], increasing its binding to eIF4E, which in turn decreases eIF4E availability, thus inhibiting RNA translation [25]. Furthermore, 4E-BP transcription is regulated by the transcription factor FOXO. Once dAkt signalling is activated, FOXO is inactivated, and 4E-BP transcription is reduced. When Akt and TOR inhibition occurs, levels and activity of 4E-BP are increased.

Herein, incremented levels of TOR expression did not show any observable impact on heart performance (Fig 2). Although cardiac TOR overexpression may be deleterious to the heart under stress at young age [9], spontaneous heart activity was not altered (Fig 2). Other experiments (e.g., direct inhibition of TOR using rapamycin in *Drosophila* larvae) also showed an unchanged heart rate [39]. The authors further remarked upon the findings of Bultinck et al. regarding rapamycin-induced SERCA inhibition [40]. Consequently, it appears to be difficult to differentiate the pharmacological actions of rapamycin on each protein involved in regulating heart physiology and performance.

On the other hand, starvation inhibits the TOR pathway and has been utilised as a strategy to explore TOR and its effectors [26, 27]. Mediated by FOXO and TOR inhibition, it could be expected that starvation allows for 4E-BP activation. Young control flies exposed to 48 hours of food deprivation showed similar effects on cardiac performance as flies exhibiting increased 4E-BP expression in the heart (i.e., increased SERCA activity and higher rates of contraction and relaxation, Figs 2 and 3). However, these starved flies also presented a reduction in transient amplitude. These somewhat paradoxical results could be attributed to a reduction in transient $Ca^{2+}$ duration produced by the augmented spontaneous heart rate. Heart rate increment as a result of starvation has been reported previously [41]. An augmented number of beats per unit of time renders it difficult to achieve high values of amplitude, even with acceleration of $Ca^{2+}$ transients being present. This option is in concordance with the fact that an increasing heart rate causes a negative staircase, as we observed in a previous report [20]. In starved flies, SERCA activity was augmented, whereas the decrease in relaxation constant and the increase in the velocity of relaxation did not reach statistical significance. These results might suggest simultaneous changes in myofilament $Ca^{2+}$ sensitivity [37]. Unexpectedly, starvation did not augment the effects of 4E-BP overexpression in this group of flies, except for an increased SERCA activity. Probably, a maximal alteration of the $Ca^{2+}$ transient might be reached already by 4E-BP overexpression, and a surplus stimulus could then be ineffective in this transgenic line.

Starvation itself, as well as inhibition of TOR pathways, affects numerous physiological processes. Food intake, a process controlled by defined mechanisms and pathways [42], is just one example. Moreover, starvation affects all tissues where dTOR is present. Reducing the function of *Drosophila* TOR e.g. also results in decreased lipid storage and glucose levels [43]. Thus, the differences observed as a result of starvation-mediated inhibition of TOR signalling, in comparison to overexpression of 4E-BP, may result from starvation-induced systemic effects.

## Proposed molecular mechanism of 4E-BP mediated effects on cardiac function and discovery of a new protein-protein interaction

We have shown that modulation of cardiac activity by 4E-BP is accompanied by distinct changes in $Ca^{2+}$ cycling. One of the affected molecular factors was the activity of the $Ca^{2+}$ ATPase of the sarcoplasmic reticulum, SERCA. Given that no direct influence of the 4E-BP and eIF4E proteins on the effectors that regulate $Ca^{2+}$ transients is known, these results suggest a possible non-canonical function of both proteins in the heart. By means of two-hybrid assays,

we were able to corroborate that SERCA interacts physically with eIF4E-4. eIF4E isoforms 1 and 2 showed a weaker interaction with SERCA (Fig 4). Based on the strong interaction as well as its confirmed expression in heart tissue (Fig 5, Table 1), we propose that predominately eIF4E-4 is relevant to cardiac calcium dynamics.

The discovery of multiple eIF4E paralogs present in a many species led to the proposal that a protein exists that performs the global and routine translation in the organisms, while the other paralogs are specialized in specific translation within a single tissue or under certain physiological conditions [5]. Thus, in the fruit fly eIF4E-1 represents the routine variant, while eIF4E-3 is testis specific [28]. We detected peptides belonging to eIF4E 1 and 2 in our mass spectra. However, modulation of 4E-BP on eIF4E-1/2 might not affect the intracellular calcium handling. Our results indicated that overexpression of TOR, the 4E-BP inhibitory kinase, did not produce changes in intracellular calcium dynamics in the individuals analyzed (Fig 2). Therefore, it is possible that changes observed in the calcium transients in flies overexpressing 4E-BP were not associated with global translation modulation, even when eIF4E-1/2 is present.

In mammals, the activation of mTOR signalling is associated with the development of cardiac hypertrophy [44, 45]. Thus, mTOR phosphorylates 4E-BP and reduces its inhibitory effect on eIF4E by stimulating protein synthesis [44]. Under normal conditions, cardiomyocytes from the adult mammalian heart have a low capacity to proliferate [46, 47], and heart protein synthesis levels are low [48]. However, there is possible translational regulation of specific proteins. Recent studies confirm the ability of 4E-BP to repress the translation of specific target mRNAs [49].

We did not measure translation or hypertrophy in the adult *Drosophila* heart. One possibility is that it behaves similarly to the mammalian heart in that overall levels of translation are not elevated. In addition, we observed that overexpression of TOR kinase did not produce changes in the calcium transient relative to wild-type flies (Fig 2). This suggests that changes observed in the calcium transient in flies overexpressing 4E-BP were not associated with global translation modulation. However, it cannot be ruled out that the translation of specific proteins is affected. As mentioned before, aging might affect translation of several proteins [29, 50]. This aspect needs to be explored further.

Based on our data, we suggest that eIF4E-4 is a cardiac specific regulator of SERCA activity and $Ca^{2+}$ handling in *Drosophila melanogaster*, with regulation being based on direct protein-protein interactions between the two factors. Of note, it has been shown that several proteins interact with and modulate the SERCA pump in mammalian cardiomyocytes. In this way, the occurrence of a SERCA regulatome has been demonstrated, consisting of proteins with canonical functions different from $Ca^{2+}$ handling, such as HAX-1 (HS-1 associated protein X-1), Hsp20 (heat shock protein 20) and SUMO1 [51]. In view of our data, we consider it likely that eIF4E-4 represents a corresponding factor in the *Drosophila* heart. We furthermore propose a functional relation between 4E-BP, eIF4E-4 and SERCA in *Drosophila* cardiac tissue, by which SERCA acts as a 4E interaction protein (4E-IP). This interaction might modulate the activity of SERCA in *Drosophila melanogaster* (Fig 6).

Further studies will be necessary to clarify the influence of aging in 4E-BP action on cardiac performance. Its modulation through upstream effectors could influence the expression levels or phosphorylation states of 4E-BP. Although these issues were not explored herein, they should be studied in the future. We did not analyze the influence of TOR on cardiac function in older flies. However, stress and aging promote protein aggregation, organelle dysfunction and oxidative processes, which eventually might lead to cell damage and impairments in tissue repair, promoting dysfunction and development of age-dependent diseases [52]. 4E-BP affects translational processes through its interaction with eIF4E-1, the ubiquitous eIF4E [53], and

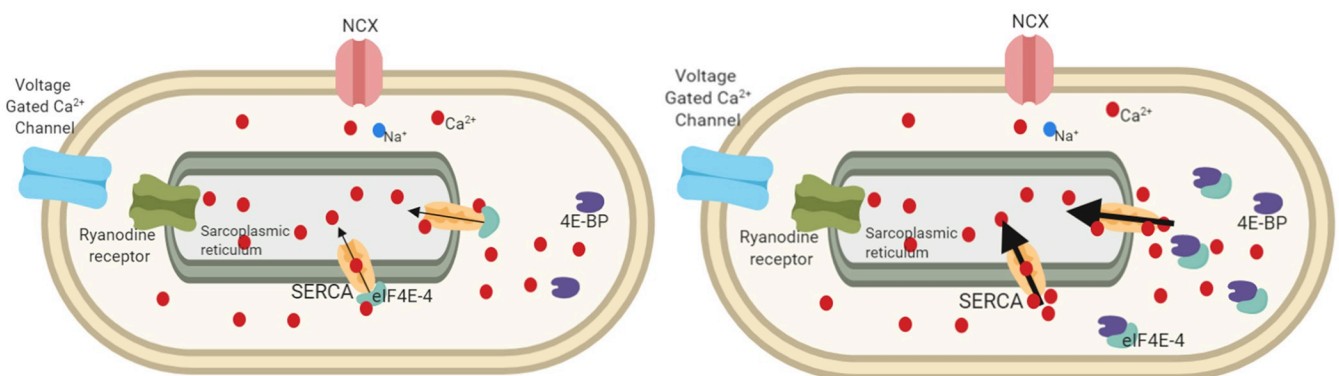

**Fig 6. Representative diagram of a wildtype cardiomyocyte and one that overexpresses 4E-BP.** This model proposes that overexpressed 4E-BP competes with SERCA for eIF4E-4, thereby reducing eIF4E-4 binding to SERCA and causing increased SERCA activity and improved cardiac performance. $Ca^{2+}$ handling-proteins are indicated: Voltage gated $Ca^{2+}$ channel, eukaryotic translation initiation factor, variant 4 (eIF4E-4), $Ca^{2+}$-ATPase (SERCA), ryanodine receptor, $Na^+/Ca^{2+}$ exchanger (NCX), calcium ion ($Ca^{2+}$), sodium ion ($Na^+$).

this interaction might alter the abundance of other proteins in the heart during aging. Summarising, the triad 4E-BP, eIF4E-4 and SERCA should be explored in more detail in the future to understand the effects of ageing in the cardiomyocyte response to variable levels of 4E-BP.

## Conclusions

The present study focuses on the relevance of 4E-BP to cardiac performance and intracellular $Ca^{2+}$ handling in *Drosophila melanogaster*. Improved cardiac function in young flies overexpressing 4E-BP appears to be associated to incremented SERCA activity. However, aging counteracts the positive influence of 4E-BP on heart function. 4E-BP is a regulatory protein of eIF4E. We observed that eIF4E-4 interacts with SERCA and is present in cardiac tissue. 4E-BP and eIF4E-4 might integrate, together with other proteins, the regulatome of SERCA, a protein, which is highly relevant to proper $Ca^{2+}$ cycling in the heart.

## Supporting information

**S1 Fig. Acquisition of the digitalized recordings and parameters measured.** A. Image taken in the microscope (left) and digitalized recording of changes in fluorescence over time, viewed with the LabChart software (right). B Left. Example of one beat that shows all parameters measured: amplitude, expressed as relative change of fluorescence between systole and diastole, normalized by the minimum of fluoresce during diastolic period (F-F0/F0). First derivative of increment and decay of the $Ca^{2+}$ transient (+ΔF/dt and -ΔF/dt). Constant of relaxation Tau. Right. Caffeine-induced calcium transient shows an increment of fluorescence and prolonged relaxation compared to the pre-caffeine $Ca^{2+}$ transients.
(TIF)

**S1 Raw image. Raw image corresponding to the gel presented in Fig 5.** Fragments of tubulin and eIF4E-4 were identified according to their expected size (*tubulin*: 182 bp; *eIF4E-4*: 166 bp).
(TIF)

**S1 Table. Analysis of SERCA activity and constant of relaxation Tau.** Average results from a subset of individuals in which Tau, as well as SERCA activity were measured. Incremented SERCA activity is associated with a relaxation acceleration. All results are expressed as mean ± SEM. * p<0.05.
(PDF)

**S2 Table. Raw data sets corresponding to parameters presented in Figs 1 to 3, measured on hearts of adult flies.** Analysis of individuals according to different ages (7 and 40 days old), strain (control, overexpression 4E-PB or TOR), and condition (i.e. starvation).
(XLSX)

## Acknowledgments

We thank Ana Clara Maldonado for her technical assistance.

## Author Contributions

**Conceptualization:** Alicia Mattiazzi, Carlos A. Valverde, Greco Hernández, Paola Ferrero.

**Data curation:** Paola Ferrero.

**Formal analysis:** Manuela Santalla, Alejandra García, Alicia Mattiazzi, Carlos A. Valverde, Ronja Schiemann, Paola Ferrero.

**Funding acquisition:** Paola Ferrero.

**Methodology:** Manuela Santalla, Alejandra García, Carlos A. Valverde, Ronja Schiemann, Achim Paululat, Greco Hernández, Heiko Meyer, Paola Ferrero.

**Project administration:** Paola Ferrero.

**Resources:** Alicia Mattiazzi, Achim Paululat, Greco Hernández, Heiko Meyer.

**Supervision:** Paola Ferrero.

**Writing – original draft:** Manuela Santalla, Alicia Mattiazzi, Carlos A. Valverde, Achim Paululat, Greco Hernández, Heiko Meyer, Paola Ferrero.

**Writing – review & editing:** Alicia Mattiazzi, Carlos A. Valverde, Achim Paululat, Greco Hernández, Heiko Meyer, Paola Ferrero.

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
