## [Decision Letter · Decision Letter 0]

14 Oct 2021

PONE-D-21-29388Interplay between SERCA, 4E-BP and eIF4-E in the Drosophila heartPLOS ONE

Dear Dr. Ferrero,

Thank you for submitting your manuscript to PLOS ONE. After careful consideration, we feel that it has merit but does not fully meet PLOS ONE’s publication criteria as it currently stands. Therefore, we invite you to submit a revised version of the manuscript that addresses the points raised during the review process. Please carefully consider all comments by the reviewer and I would expect you to address each of the suggestions for improvement in your revision.

We look forward to receiving your revised manuscript.

Kind regards,

Thomas Preiss, PhD

Academic Editor

PLOS ONE

Journal Requirements:

[This work was supported by PICT 2014-2549 ANPCyT and SIB 2015 to P.F, by PICT 2014-2524 and PIP 0350 to AM and by grants from the Deutsche Forschungsgemeinschaft to A.P. and H.M (SFB 944, Physiology and dynamics of cellular microcompartments). G.H. was supported by internal funding of the National Institute of Cancer (Instituto Nacional de Cancerología, INCan), Mexico. A National Council of Science and Technology (CONACyT) PhD fellowship was awarded to A.G. (Nr. 436200). A.G. is part of the Program on Biological Sciences, UNAM (Doctorado en Ciencias Biológicas de la Universidad Nacional Autónoma de México).]

 [This work was supported by PICT 2014-2549 ANPCyT and SIB 2015 to P.F, by PICT 2014-2524 and PIP 0350 to AM and by grants from the Deutsche Forschungsgemeinschaft to A.P. and H.M (SFB 944, Physiology and dynamics of cellular microcompartments). G.H. was supported by internal funding of the National Institute of Cancer (Instituto Nacional de Cancerología, INCan), Mexico. A National Council of Science and Technology (CONACyT) PhD fellowship was awarded to A.G. (Nr. 436200). A.G. is part of the Program on Biological Sciences, UNAM (Doctorado en Ciencias Biológicas de la Universidad Nacional Autónoma de México).The funders had no role in study design, data collection and analysis, decision to publish, or preparation of the manuscript.]

Reviewers' comments:

Reviewer's Responses to Questions

**Comments to the Author**

1. Is the manuscript technically sound, and do the data support the conclusions?

Reviewer #1: Partly

2. Has the statistical analysis been performed appropriately and rigorously? 

Reviewer #1: Yes

3. Have the authors made all data underlying the findings in their manuscript fully available?

Reviewer #1: Yes

4. Is the manuscript presented in an intelligible fashion and written in standard English?

Reviewer #1: Yes

5. Review Comments to the Author

Reviewer #1: Santalla et all showed that there is a non-canonical role of the eukaryotic translation factor 4E-BP in modulating the heart function through SERCA.

They showed that an overexpression of 4E-BP augmented calcium uptake and release in the freshly dissected fly heart. Authors attributed this effect to the activation of SERCA and the direct interaction of SERCA to eIF-4E proteins. Experiments are coherently presented and overall support their findings.

Since the interaction of SERCA and eIF4Es was not expected, more supporting evidence is needed to establish the functional/physical connection.

I recommend adding either or both of the following experiments:

1. A further dissection of the SERCA domain by Y2H to narrow down the interaction domain would validate the specific interaction between SERCA and eIF4Es. In the presented data, eIF4E-4 gave all positive results. Therefore, a simple technical error cannot be excluded.

2. Testing the effect of depleting eIF4E-4 on the heart contraction/relaxation to see if it has the same effect as 4E-BP overexpression

Minor comments:

1. Please state that different "isoforms" of eIF4E are encoded by different genes. "Isoforms" imply transcripts coming from a single gene.

2. Figure panels benefit from having more captions and labels. For example, labelling "contraction" and "relaxation" on panels C and D in Figure 1 would help.

3. Please describe how exactly the parameters of calcium dynamics were calculated, for example, by showing examples of the raw data (including caffeine experiment). I understand what each parameter captures conceptually. But, windows and thresholds taken for calculations are also important to evaluate the robustness of the experiments.

4. In Figure 1, the relaxation decay constant tau did not change much upon over expression of 4E-BP in 7-day old flies while SERCA activity increased. Does this mean that the activity of Na/Ca exchanger decreased upon overexpression of 4E-BP? Is this likely?

5. Based on the tissue expression atlas available from the FlyBase, eIF4E-1/2 gene is most abundantly expressed among eIF4E genes in most tissues except that the testes abundantly express eIF4E-3 and eIF4E-4 genes. Therefore, isn't it possible that eIF4E-1/2 is more relevant as a potential effector of 4E-BP overexpression despite less strongly interacting with SERCA in Y2H?

Did you find peptides of eIF4E-1/2 in the mass-spec data?

6. At least part of the effects of 4E-BP overexpression could be due to changes in translation, indirectly affecting the heart function. Could you comment on this possibility?

6. PLOS authors have the option to publish the peer review history of their article (what does this mean?). If published, this will include your full peer review and any attached files.

Reviewer #1: **Yes: **Rippei Hayashi

---

## [Author Response · Author response to Decision Letter 0]

17 Mar 2022

Following, we add the response to the requirements and questions:

The file has been modified according to the style templates.

Funding information has been removed from the manuscript. 

3. PLOS ONE now requires that authors provide the original uncropped and unadjusted images underlying all blot or gel results reported in a submission’s figures or Supporting Information files. 

We add the raw gel image in Supporting Information file (S2 fig).

The data are not a core part of the research in this study therefore the phrase has been removed. 

References were revised and there are not retracted articles.

Reviewers' comments:

Reviewer's Responses to Questions

Comments to the Author

1. Is the manuscript technically sound, and do the data support the conclusions?

Reviewer #1: Partly

2. Has the statistical analysis been performed appropriately and rigorously?

Reviewer #1: Yes

3. Have the authors made all data underlying the findings in their manuscript fully available?

Reviewer #1: Yes

4. Is the manuscript presented in an intelligible fashion and written in standard English?

Reviewer #1: Yes

5. Review Comments to the Author

Reviewer #1: Santalla et all showed that there is a non-canonical role of the eukaryotic translation factor 4E-BP in modulating the heart function through SERCA.

They showed that an overexpression of 4E-BP augmented calcium uptake and release in the freshly dissected fly heart. Authors attributed this effect to the activation of SERCA and the direct interaction of SERCA to eIF-4E proteins. Experiments are coherently presented and overall support their findings.

Since the interaction of SERCA and eIF4Es was not expected, more supporting evidence is needed to establish the functional/physical connection.

I recommend adding either or both of the following experiments:

1. A further dissection of the SERCA domain by Y2H to narrow down the interaction domain would validate the specific interaction between SERCA and eIF4Es. In the presented data, eIF4E-4 gave all positive results. Therefore, a simple technical error cannot be excluded.

2. Testing the effect of depleting eIF4E-4 on the heart contraction/relaxation to see if it has the same effect as 4E-BP overexpression

We proceed to carry out the experiment suggested by the reviewer on point 1. A new figure 4 is added to the manuscript. This figure includes the analysis of two shorter fragments of SERCA, namely amino acids 1-250 and 200-250. As we detailed in Results Section in the new version of this manuscript, the SERCA constructs 1-250 and 51-250, but not construct 200-250, interacted with eIF4E-4. This result suggests that the SERCA eIF4E-binding motif is located within region encompassing amino acids 1-200. 

Please, see the new figure 4. Pages 13 and 14 contains the rewritten results highlighted (lines 364 to 373). Figure legend was modified according to the annexed experiment (lines 375-383). In the section method, page 6 shows the description of the experiment.

Minor comments:

1. Please state that different "isoforms" of eIF4E are encoded by different genes. "Isoforms" imply transcripts coming from a single gene.

All sentences were revised and reformulated according to the reviewer suggestion. We conserved the term isoform for referring to eIF4E-1 and 2 with results from alternative splicing.

2. Figure panels benefit from having more captions and labels. For example, labelling "contraction" and "relaxation" on panels C and D in Figure 1 would help.

We greatly thank the reviewer for the suggestion. Labeling has been improved in panels C, D and E in Figures 1, 2 and 3. 

3. Please describe how exactly the parameters of calcium dynamics were calculated, for example, by showing examples of the raw data (including caffeine experiment). I understand what each parameter captures conceptually. But, windows and thresholds taken for calculations are also important to evaluate the robustness of the experiments.

We added information in the section methods as follow.

“The Ca2+ transients were recorded during 30 seconds resulting in a 1024-pixel image. The images obtained were interpreted to graph sequentially in time the intensity of fluorescence (S1 Fig), mediating a customized algorithm using the Jupiter notebook program, based on Python programming language, with Matplotlib and NumPy libraries from Anaconda (Balcazar et al 2018). Values expressed as relative change of fluorescence along time, calculated according to the formula: Fmax-F0/F0 and expressed in arbitrary units of fluorescence (AU), were analyzed with LabChart software (AD Instruments, CO, USA). Measurements included: peak Ca2+ transient amplitude (Fmax-F0/F0) (AU), maximal rates of fluorescence increasing and decreasing (+dΔF/dt) (sec), (-dΔF/dt) (sec). Relaxation was measured by calculating the Tau constant (in seconds) of the transient exponential decay of Ca2+ (S1 Fig).” Please, see pages 5 and 6, lines 135 to 149).

Moreover, we included one supporting information figure that shows an image of the fluorescence, a digitalized recording, and the measured parameters.

4. In Figure 1, the relaxation decay constant tau did not change much upon over expression of 4E-BP in 7-day old flies while SERCA activity increased. Does this mean that the activity of Na/Ca exchanger decreased upon overexpression of 4E-BP? Is this likely?

Thank you for this important question. Actually, the first bar of Figure 1 was drawn incorrectly. We apologize for this error. However, and as the reviewer indicated this difference, although greater than before, did not reach statistical difference. The activity of the NCX did not decrease. The reason for the lack of correlation between the time constant of transient decay in 4E-BP flies and the increase in SERCA activity might result from the comparison of independent preparations. To test this possibility and with the aim of clarifying this point and verifying a direct impact of SERCA activity on calcium transient decay, we analyzed the constant of relaxation Tau for a subset of data in which SERCA activity were simultaneously measured. An increase in SERCA activity was correlated with an acceleration of relaxation (S1 table). 

However, an action of 4EBP on other targets associated with intracellular calcium handling could not be ruled out. For example, it has been shown in cell lines that 4EBP modulates mitochondrial biogenesis and activity (Morita et al, 2013). Mitochondria are a slow calcium removal system in vertebrates (Eisner et al, 2017), but their contribution to intracellular calcium removal in the Drosophila heart is unknown.

Moreover, intracellular calcium handling is influenced by the sensitivity of contractile proteins to calcium. During the relaxation phase, calcium must be released from troponin C to promote dissociation of actin and myosin filaments. If the calcium sensitivity of contractile proteins is incremented, contractility also increases, but relaxation may be prolonged because calcium dissociates more slowly from protein C (Chung et al, 2016). Although we did not explore this aspect, changes in the sensitivity of contractile proteins to calcium could counteract the relaxant effect produced by increased SERCA activity in the analyzed individuals (Fig 1E).

These aspects were incorporated in results (page 11, lines 291 to 295) and discussion section (pages 16 and 17 lines 447 to 462). S1 table is provided in the supporting information file.

5. Based on the tissue expression atlas available from the FlyBase, eIF4E-1/2 gene is most abundantly expressed among eIF4E genes in most tissues except that the testes abundantly express eIF4E-3 and eIF4E-4 genes. Therefore, isn't it possible that eIF4E-1/2 is more relevant as a potential effector of 4E-BP overexpression despite less strongly interacting with SERCA in Y2H? Did you find peptides of eIF4E-1/2 in the mass-spec data?

We detected peptides belonging to eIF4E 1/2 in our mass spectra. However, modulation of 4E-BP on eIF4E1/2 might not affect the intracellular calcium handling. Our results indicate that overexpression of TOR, the 4E-BP inhibitory kinase, did not produce changes in intracellular calcium dynamics in the individuals analyzed (Fig. 2). Therefore, it is possible that changes observed in the calcium transient in flies overexpressing 4E-BP, were not associated with global translation modulation even when eIF4E1/2 is present.

This was added to the discussion section. Please, see page 20 lines 551 to 558.

6. At least part of the effects of 4E-BP overexpression could be due to changes in translation, indirectly affecting the heart function. Could you comment on this possibility?

We agree with the reviewer. The following paragraphs have been incorporated in the discussion section. Please, see pages 20- 21. Lines 567 to 575.

In mammals, the activation of mTOR signalling is associated with the development of cardiac hypertrophy (Kemi et al., 2008; Maillet et al., 2013). Thus, mTOR phosphorylates 4E-BP and reduces its inhibitory effect on eIF4E by stimulating protein synthesis (Kemi et al., 2008). Under normal conditions, cardiomyocytes from the adult mammalian heart have a low capacity to proliferate (Soonpaa et al., 1996, Bergmann et al., 2015), and heart protein synthesis levels are low (Gray and Gray 2017). However, there is possible translation regulation of specific proteins. Recent studies confirm the ability of 4E-BP to repress the translation of specific target mRNAs (Jin et al, 2020).

We did not measure translation or hypertrophy in the adult Drosophila heart. One possibility is that it behaves similarly to the mammalian heart in that overall levels of translation are not elevated. In addition, we observed that overexpression of TOR kinase did not produce changes in the calcium transient relative to wild-type flies (Fig. 2). This suggests that changes observed in the calcium transient in flies overexpressing 4E-BP were not associated with global translation modulation. However, there cannot be ruled out that the translation of specific proteins is affected. As mentioned before, aging might affect translation of several proteins (Basisty et al, 2018). This aspect must be further explored.

---

## [Decision Letter · Decision Letter 1]

4 Apr 2022

Interplay between SERCA, 4E-BP and eIF4-E in the Drosophila heart

PONE-D-21-29388R1

Dear Dr. Ferrero,

We’re pleased to inform you that your manuscript has been judged scientifically suitable for publication and will be formally accepted for publication once it meets all outstanding technical requirements.

Kind regards,

Thomas Preiss, PhD

Academic Editor

PLOS ONE

Additional Editor Comments (optional):

Reviewers' comments:

Reviewer's Responses to Questions

**Comments to the Author**

1. If the authors have adequately addressed your comments raised in a previous round of review and you feel that this manuscript is now acceptable for publication, you may indicate that here to bypass the “Comments to the Author” section, enter your conflict of interest statement in the “Confidential to Editor” section, and submit your "Accept" recommendation.

Reviewer #1: All comments have been addressed

2. Is the manuscript technically sound, and do the data support the conclusions?

Reviewer #1: Yes

3. Has the statistical analysis been performed appropriately and rigorously? 

Reviewer #1: Yes

4. Have the authors made all data underlying the findings in their manuscript fully available?

Reviewer #1: Yes

5. Is the manuscript presented in an intelligible fashion and written in standard English?

Reviewer #1: Yes

6. Review Comments to the Author

Reviewer #1: The authors carefully addressed all of the reviewer's comments. The manuscript is ready to be published.

7. PLOS authors have the option to publish the peer review history of their article (what does this mean?). If published, this will include your full peer review and any attached files.

Reviewer #1: **Yes: **Rippei Hayashi

---

## [Editor Report · Acceptance letter]

28 Apr 2022

PONE-D-21-29388R1 

Interplay between SERCA, 4E-BP, and eIF4E in the *Drosophila* heart 

Dear Dr. Ferrero:

I'm pleased to inform you that your manuscript has been deemed suitable for publication in PLOS ONE. Congratulations! Your manuscript is now with our production department. 

Kind regards, 

on behalf of

Prof Thomas Preiss 

Academic Editor

PLOS ONE